# Uptake of cervical cancer screening and its associated factors among women of reproductive age in Kericho County

Joel Wanzala<sup>©</sup>, Calvince Otieno Anino ©*

Public Health Department, University of Kabianga, Kericho, Kenya

© These authors contributed equally to this work.
¤ Current address: Department of Public Health, University of Kabianga, Kericho, Kenya
* calvinceanino@gmail.com

## Abstract

Cervical cancer is a public health issue among reproductive-aged women worldwide. It is the second most common cancer among females and contributes to 12.9% of new cancer cases and 11.84% of all cancer deaths annually. Early detection and treatment can prevent and cure the disease. Screening among women has gained global attention since it's a crucial step in early detection. This study investigated the level of uptake of cervical cancer screening and factors associated with the cervical cancer screening among women aged 18–49 years in Bureti Constituency, Kericho County. The study adopted an institution-based cross-sectional study design. The study used systematic random sampling technique to select 328 women aged 18–49 who attended maternal and child health clinic at Kapkatet Sub-County Hospital during study period. Data was collected using structured questionnaires, including sociodemographic, awareness and knowledge and screening uptake sections. The data was analyzed using SPSS version 26.0. Descriptive analyses were performed to all the variables. Binary logistic regression analyses were conducted to assess factors associated with cervical cancer screening. The uptake of cervical cancer screening was 16.2% (n = 53, 95% CI: 10.5–23.4). We found that being 34 years and above (aOR = 1.43, 95% CI: 1.11–2.52, p < 0.001), having a primary education (aOR = 3.20, 95% CI: 1.50–7.00, p = 0.003), being self-employed (aOR = 5.20, 95% CI: 2.40–10.70, p < 0.001), or unemployed (aOR =7.50, 95% CI: 2.45–25.00, p < 0.001), and having a family history of cervical cancer (aOR = 2.10, 95% CI: 0.80–6.00, p = 0.015) significantly increased the odds of screening uptake. In contrast, earning an income of Ksh. 2,001–5,000 (aOR = 0.38, 95% CI: 0.10–0.90, p = 0.016) or> Ksh. 10,000 (aOR = 0.15, 95% CI: 0.06–0.33, p < 0.001) significantly decreased the odds of screening. Additionally, poor knowledge on risk factors (aOR = 0.50, 95% CI: 0.28–0.65, p < 0.001), signs and symptoms (aOR = 0.40, 95% CI: 0.10–0.70, p < 0.001), and groups at risk (aOR = 0.35, 95% CI: 0.15–0.60, p < 0.001) were all associated with

**Data availability statement:** All relevant data are within the manuscript and its Supporting Information files.

**Funding:** The author(s) received no specific funding for this work.

**Competing interests:** The authors have declared that no competing interests exist.

lower odds of screening. Uptake of cervical cancer screening was influenced by multiple factors including socio-demographics and economic factors, awareness, and knowledge related factors. Thus, there is urgent need for intensified health education to enhance awareness and knowledge of women on cervical cancer and its screening.

## Introduction

Cervical cancer is a prevalent non-communicable and public health issue among reproductive-aged women worldwide [1], with approximately 660,000 women developing the disease annually [2,3]. With early detection and treatment, it is a preventable and curable disease [4]. Sub-Saharan Africa has the highest regional incidence and mortality, with Eastern Africa, Southern Africa, and central Africa having the highest rates [5,6]. Cervical cancer rates are disproportionately high in developing countries, with low Human Development Index (HDI) and poverty rates responsible for over 52% of the global variance in mortality [7]. The absence of effective cervical screening has led to a rapid elevation in premature cervical cancer mortality in recent years. In Kenya, cervical cancer is one of the major causes of cancer-related deaths and the second most common cancer among females, contributing to 12.9% of new cancer cases and 11.84% of all cancer deaths annually [7,8]. It is the leading cause of all cancer deaths in Kenya, with over 3,200 deaths in 2020 [2]. Public health programs, such as cervical cancer screening of women for precancerous changes, treatment, and follow-up care at the early stages of the disease, are reported to be more significant [9].

Studies by D'Augè et al. [9] and Singh et al. [10] have identified cervical cancer screening as the best approach towards achieving significant reduction in the incidence, morbidity, and mortality of cervical cancer. It's therefore important for all women within the reproductive age bracket (15–49 years) to have cervical cancer screenings to detect any abnormal cells. As reported by Teklehaimanot et al. [11] if cervical cancer is detected early, timely treatment should be initiated to avoid the precancerous cells from developing into tumors or carcinoma. Experts concur that cervical cancer screening should be done via visual inspection of the cervix with lugol's iodine (VILI), a papanicolaou (PAP) smear, and a visual inspection of the cervix with acetic acid (VIA) [12]. Once screened negative, it is advised that rescreening should be done after five years [13]. However, these screening processes have been reported as laborious and could potentially be a reason to shun screening [14]. This is a concern given that there has been a rise in the number of new cases from 12.9% in 2018 to about 19.7% in 2020 [2]. Besides, Ng'ang'a et al. [15] reported low uptake of screening and screening services of about 16% of Kenyan women within reproductive age seeking for screening services and about a quarter of the health facilities offering screening services between 2015 and 2018. This is alarming given that Kenya has been implementing a national screening program in the past 10 years [3]. Based on Gebreegziabher's report, the high level of awareness among Kenyan

women suggests that cancer screening uptake should be high [16]. While awareness of cervical cancer is reportedly high among Kenyan women, the reasons for the consistently low screening uptake remain largely unaddressed. Therefore, this study aimed to assess the prevalence and factors associated with cervical cancer screening among women aged 18–49 years in Bureti Constituency, Kericho County.

## Materials and methods

### Study design and setting

The study adopted an institution-based cross-sectional study design. The study was carried out among women aged 18–49 years who attended Kapkatet Sub-County Hospital in Kericho County during the study period. The hospital is a major healthcare facility in Bureti Constituency, in a rural setting. The inclusion criteria comprised consented participants, women aged 18–49 who attended Kapkatet Sub-County Hospital. Women who were severely ill; unable to speak or hear; and lived in the study area for less than 6 months were excluded from the study.

### Variables

This study investigated uptake and factors such as sociodemographic and knowledge related factors that influence cervical cancer screening utilization. Sociodemographic factors included age, religion, marital status, education attainment, employment status, household monthly income, age at first marriage, family history of cervical cancer, parity, sexual partners, and distance to the nearest health facility. Knowledge and uptake of cervical cancer included variables such as awareness of cervical cancer, patient, screening and screened persons, and knowledge on risk factors, signs and symptoms, preventive measures and groups at risk. Additionally, dependent variable was uptake of cervical cancer screening.

### Sample size and sampling technique

Sample size was determined using Fischer's formula as described by Fisher et al. [17] and assumed uptake of 50% and error margin of 5%. The study achieved response rate of 85.4% (n = 328). Systematic random sampling technique was used to select 328 women aged 18–49 years attending mother and child health clinic. Eligible women were selected based on their arrival at the facility during study period. The first eligible woman was selected randomly while the rest were selected based on their arrival at predetermined sampling interval of 2nd woman who arrives at the clinic. This continued for a period of 3 months, with keen to double selection of already enrolled participants.

### Data collection tool and procedure

This study collected data from January to March 2024 using structured questionnaires administered by trained and experienced research assistants under supervision of principal researcher. The questionnaires were pretested, customized, standardized and adapted from various studies [18,19]. The questionnaire had three sections; sociodemographic characteristics, awareness and knowledge and screening uptake. The sociodemographic section entailed 11 items and asked questions related to participant's age, religion, marital status, education attainment, employment status, household monthly income, age at first marriage, family history of cervical cancer, parity, sexual partners, and distance to the nearest health facility. Awareness and knowledge section focused on questions that focused on awareness of cervical cancer, patient, screening and screened persons, and knowledge on risk factors, signs and symptoms, preventive measures and groups at risk. Additionally, cervical cancer screening uptake section was assessed through a dichotomous question (recorded as yes/no responses), that focused to ascertain whether or not they had ever undergone screening. Informed consent was sought from each participant prior enrollment to the study; only who gave and met inclusion criteria were included.

## Data analysis

Collected data was checked for completeness, cleaned, coded, and analyzed using Statistical Package for Social Science (SPSS) software version 26.0. Descriptive analyses were done for all variables, and summarized using frequencies and percentages. Knowledge related factors were transformed into one single variable scored as poor or good. Those who scored 50% or above were considered to have good knowledge, while those who scored below 50% were considered to have poor knowledge. This categorization was based on predefined cut-off scores derived from a scoring system developed through consensus among researchers familiar with the subject matter. Binary logistic regression analyses were conducted to assess association between the sociodemographic and knowledge related factors, and screening uptake. Results summarized using adjusted odds ratio, 95% confidence interval, and p-values. A p value of < 0.05 was considered significant.

## Ethical considerations

The study obtained ethical approval from the University of Kabianga-Kapkatet Institutional Research Ethical Committee (ISERC/2022/0154). Ethical clearance was also acquired from the Kenyan National Commission for Science, Technology, and Innovation (NACOSTI/P/23/25008). Permission was sought from the medical superintendent of the hospital. All recruited participants were informed about the objectives of the research. Written informed consent was obtained from each study participant. The names of respondents were not written on the questionnaire, but instead, a unique code was used to identify each participant and all information that was obtained from the health institution and respondents was kept strictly confidential. The written consent forms were kept in a lockable cabinet which could only be accessed by the investigators.

## Results

### Socio-demographic characteristics of study participants

The study included a total of 328 women of reproductive age. out of 328 respondents, 48 (33.1%) were aged 18–24 years; the majority were Christians (97.2%), married (50.3%), had attained secondary education (34.5%), reported business as their main source of income (39.3%), and earned less than Ksh. 2,000 (46.9%) as shown in Table 1. Most of the respondents had two to four children (53.8%), had no multiple sex partners (56.6%), and had no family history of cervical cancer (98.6%). Out of 96 respondents who have ever been married, 63.5% were first married at the age of 18 years or below.

### Association between socio-demographics and economics, awareness and knowledge related factors to uptake of cervical cancer screening

Association between socio-demographic and economics characteristics, awareness and knowledge related factors to cervical cancer screening uptake were carried out using both chi-square and logistic regression modelling. The uptake of cervical cancer screening was 16.2% (n = 53, 95% CI: 10.5–23.4). All the respondents who screened were aware of cervical cancer, 67.8% knew at least one cervical cancer patient, all had heard cervical cancer screening, and 92.5% knew at least someone who had screened as shown in Table 2. Additionally, among those who screened, majority had good knowledge on risk factors of cervical cancer (88.7%), its signs and symptoms (69.8%), and groups at risk of cervical cancer. However, more than half (54.7%) of screened women had poor knowledge on preventive measures of cervical cancer. All the awareness and knowledge related factors were associated with cancer screening uptake.

   Logistic regression model was carried out to identify the independent predictors of cervical cancer screening with the aORs indicating the strength and direction of the association as shown in Table 3. Women aged ≥ 34 had a significantly higher odds of taking the cervical cancer screening compared to those aged 18–24 (aOR = 1.43, p < 0.001). In terms of level of education, having a primary school education was significantly associated with 3.20 times higher odds

**Table 1. Socio-demographic and economics characteristics of study respondents.**

| Variable | Frequency (N = 328) | Percentage % |
|---|---|---|
| **Age** | | |
| 18–24 | 115 | 33.1 |
| 25–34 | 138 | 26.9 |
| ≥ 35 | 75 | 30.0 |
| **Religion** | | |
| Christianity | 319 | 97.3 |
| Muslim | 9 | 2.7 |
| **Marital status** | | |
| Married | 162 | 49.4 |
| Unmarried | 166 | 50.6 |
| **Level of education** | | |
| No formal | 26 | 7.9 |
| Primary | 102 | 31.1 |
| Secondary | 111 | 33.3 |
| Tertiary | 89 | 27.1 |
| **Employment status** | | |
| Formally employed | 44 | 13.4 |
| Self – employed | 237 | 72.3 |
| Unemployed | 47 | 14.3 |
| **Monthly household income** | | |
| <Ksh. 2, 000 | 153 | 46.6 |
| Ksh. 2, 001–5, 000 | 78 | 23.8 |
| Ksh. 5, 001–10, 000 | 53 | 16.2 |
| > Ksh. 10, 000 | 44 | 13.4 |
| **Parity** | | |
| None | 69 | 21.0 |
| 1–3 | 189 | 57.0 |
| ≥ 4 | 70 | 21.4 |
| **Multiple sex partners** | | |
| Yes | 141 | 43.0 |
| No | 187 | 57.0 |
| **Family history of cervical cancer** | | |
| Yes | 4 | 1.2 |
| No | 324 | 98.8 |
| **Age at first marriage (n = 212)** | | |
| 18 Years or Below | 134 | 63.2 |
| Above 18 years | 78 | 36.8 |
| **Distance from health facility** | | |
| Less than 4 KM | 181 | 55.2 |
| More than 4 KM | 147 | 44.8 |

of screening uptake compared to having no formal education (aOR = 3.20, p = 0.003). Regarding employment status, both being self-employed and unemployed were significantly associated with a much higher odds of screening uptake compared to being formally employed (aOR = 5.20 and 7.50, respectively, both p < 0.001). For monthly household

**Table 2. Association between awareness and knowledge related factors, and uptake of cervical cancer screening.**

| Variable | Screened n (%) n = 53 | Not screened n (%) n = 275 | Frequency N (%) | P |
|---|---|---|---|---|
| **Aware of cervical cancer** | | | | |
| Yes | 53 (19.4%) | 220 (80.6%) | 273 (83.2) | 0.031 |
| No | 0 (0.0%) | 55 (100.0%) | 55 (16.8) | |
| **Know cervical cancer patient** | | | | |
| Yes | 37 (39.8%) | 56 (60.2%) | 93 (28.3) | 0.044 |
| No | 16 (6.8%) | 219 (93.2%) | 235 (71.7) | |
| **Ever heard of cervical cancer screening** | | | | |
| Yes | 53 (35.1%) | 98 (64.9%) | 151 (46.0) | 0.029 |
| No | 0 (0.0%) | 177 (100.0%) | 177 (54.0) | |
| **Know someone screened** | | | | |
| Yes | 49 (51.6%) | 46 (48.4%) | 95 (29.0) | < 0.001 |
| No | 4 (1.7%) | 229 (98.3%) | 233 (71.0) | |
| **Knowledge on risk factors** | | | | |
| Good knowledge | 47 (51.6%) | 44 (48.4%) | 91 (27.7) | < 0.001 |
| Poor knowledge | 6 (2.5%) | 231 (97.5%) | 237 (72.3) | |
| **Knowledge on signs and symptoms** | | | | |
| Good knowledge | 37 (53.6%) | 32 (46.4%) | 69 (21.0) | 0.036 |
| Poor knowledge | 16 (6.2%) | 243 (93.8%) | 259 (79.0) | |
| **Knowledge on prevention measures** | | | | |
| Good knowledge | 24 (34.3%) | 45 (65.7%) | 69 (21.0) | < 0.001 |
| Poor knowledge | 29 (11.2%) | 230 (88.8%) | 259 (79.0) | |
| **Knowledge on groups at risk** | | | | |
| Good knowledge | 33 (28.0%) | 85 (72.0%) | 118 (36.0) | 0.018 |
| Poor knowledge | 20 (9.5%) | 190 (90.5%) | 210 (64.0) | |

*p* refers to p – value.

income, having an income greater than Ksh. 10,000 was significantly associated with 0.15 times lower odds (aOR = 0.15, p < 0.001). A family history of cervical cancer was significantly associated with 2.10 times higher odds (aOR = 2.10, p = 0.015). Factors associated with lower odds of cervical cancer screening included, being married after age 18 (aOR = 0.70, p < 0.001), knowing a cervical cancer patient (aOR = 0.60, p = 0.001), and having heard of cervical cancer screening (aOR = 0.70, p = 0.038). Finally, knowledge on risk factors, signs and symptoms, and groups at risk were all significantly associated with a lower adjusted odds of the screening uptake (aOR = 0.50, 0.40, and 0.35 respectively, all p < 0.001).

## Discussion

Despite the fact that cervical cancer screening reduces the incidence and mortality rates associated with cervical cancer, women in most developing nations have reported significantly low screening utilization, particularly in Kenya. The results of this study revealed that the prevalence of cervical cancer screening was 16.2%. This finding is consistent studies by Ng'ang'a et al. and Ngetich et al. which reported cervical cancer screening uptake of 14.4% and 15%, respectively [15,20]. Similarly, a study in Nigeria reported that only 16.2% of antenatal clinic attendees had screened for cervical cancer [21]. The Kenyan Demographic and Health Survey 2022 report showed a national prevalence of cervical cancer screening of 16.7%, which is lower than projected 90-70-90 target by 2030, to achieve 70% cervical cancer screening uptake. Lower screening uptake might be attributed to inadequate awareness campaigns and stimulating factors that can promote the

**Table 3. Logistic regression analysis of association between socio-demographic, economics, awareness and knowledge related factors, and uptake of cervical cancer screening.**

| Variable | aOR | 95% CI | *p* |
|---|---|---|---|
| **Age group** | | | |
| 18–24 (ref) | 1.00 | – | – |
| 25–34 | 0.44 | 0.17–1.14 | 0.09 |
| ≥ 34 | 1.43 | 1.11–2.52 | <0.001 |
| **Level of education** | | | |
| No formal (ref) | 1.00 | – | – |
| Primary | 3.20 | 1.50–7.00 | 0.003 |
| Secondary | 2.50 | 0.60–4.90 | 0.051 |
| Tertiary | 1.75 | 0.90–3.72 | 0.060 |
| **Employment status** | | | |
| Formally employed (ref) | 1.00 | – | – |
| Self-employed | 5.20 | 2.40–10.70 | <0.001 |
| Unemployed | 7.50 | 2.45–25.00 | <0.001 |
| **Monthly household income** | | | |
| <Ksh. 2,000 (ref) | 1.00 | – | – |
| Ksh. 2,001–5,000 | 0.38 | 0.10–0.90 | 0.016 |
| Ksh. 5,001–10,000 | 0.70 | 0.25–1.90 | 0.56 |
| > Ksh. 10,000 | 0.15 | 0.06–0.33 | <0.001 |
| **Family history of cervical cancer** | | | |
| No (ref) | 1.00 | – | – |
| Yes | 2.10 | 0.80–6.00 | 0.015 |
| **Age at first marriage** | | | |
| ≤ 18 years (ref) | 1.00 | – | – |
| > 18 years | 0.70 | 0.05–0.97 | <0.001 |
| **Know cervical cancer patient** | | | |
| Yes (ref) | 1.00 | – | – |
| No | 0.60 | 0.33–0.85 | 0.001 |
| **Ever heard of cervical cancer screening** | | | |
| Yes (ref) | 1.00 | – | – |
| No | 0.70 | 0.25–0.90 | 0.038 |
| **Know someone screened** | | | |
| Yes (ref) | 1.00 | – | – |
| No | 0.85 | 0.38–0.92 | 0.011 |
| **Knowledge on risk factors** | | | |
| Good knowledge (ref) | 1.00 | – | – |
| Poor knowledge | 0.50 | 0.28–0.65 | <0.001 |
| **Knowledge on signs and symptoms** | | | |
| Good knowledge (ref) | 1.00 | – | – |
| Poor knowledge | 0.40 | 0.10–0.70 | <0.001 |
| **Knowledge on groups at risk** | | | |
| Good knowledge (ref) | 1.00 | – | – |
| Poor knowledge | 0.35 | 0.15–0.60 | <0.001 |

***p* refers to p – value, aOR refers to adjusted odds ratio, ref refers to reference category, and CI refers to confidence interval.**

uptake of screening services. Therefore, integration of screening services at the community level and routine screening programs at the hospital are vital to boosting uptake of screening services. This study revealed that majority of the women who screened were aware of cervical cancer and its screening. Despite having had good knowledge on risk factors of cervical cancer, its signs and symptoms and groups at risk, they had poor knowledge on essential preventive and management measures of cervical cancer. Similarly, awareness of cervical cancer and its screening have been reported among screened women [22–24]. Lower knowledge on preventive and management strategies such as importance role of screening in early detection of cervical cancer, is common among majority of screening eligible women in Africa [25–27]. While intensification of community health education campaigns focuses to raise awareness on cervical cancer and its screening, it's critical to emphasize on need for screening, and other available essential preventive and management options.

The study revealed that socio-demographic factors such as age, level of education, employment status, household monthly income, family history of cervical cancer, and age at first marriage were significantly associated with cervical cancer screening. Women aged 34 years and above, and attained primary education were more likely to be screened compared to counterparts aged 18–24 years and had no formal education, respectively. This result was consistent with studies that reported that young women had lower likelihood to uptake cervical cancer screening compared to older women [28,29]. This might be attributed by perceived lower risk and cancer related morbidity and mortality among women of younger age groups, which hinder their intention and screening practices. Similarly, a study conducted by Ng'ang'a et al. reported that higher educational attainment was significantly associated with increased odds of screening [15]. This is attributed to more exposure to information regarding cervical cancer and its screening. Inclusion women health in education curriculum, especially cervical cancer studies, promote utilization of screening and health seeking practices due to increased awareness and knowledge. Self – employed and unemployed women had significantly higher odds of being screened compared to employed. This result was consistent with a study by Alie et al. [28] and Woldetsadik et al. [30], which attributed this to fixed and rigid work schedules. Fixed work schedules and inadequate leave period might also hinder healthcare seeking behaviour among employed women. Family history of cervical cancer was significantly associated with higher likelihood of screening. This result was supported by findings from northern Ethiopia and Kenya [18,29,31]. This can be explained by the fact that women with family history have been exposed to information about cervical cancer screening and are at risk of its development, making them more likely to be screened. This study also revealed that higher household income and marriage at the age of 18 years or more was significantly associated with lower odds of being screened. These results were in contrary with studies that found increased likelihood of screening uptake among women with the highest household monthly income compared to those from low-income backgrounds [6,32]. This difference might be attributed to subsidized or free screening services campaign in the country aimed to promote early detection of cervical cancer through timely screening, its management and reduced morbidity and mortality rates, particularly among the economically disadvantaged population. Concurring with current study's finding, study by Desta et al. [33] reported that early marriage was significantly associated with increased odds of cervical cancer screening uptake. This might be associated with early sexual debuts and higher fertility rates, hence increased contact with healthcare services for reproductive health.

Additionally, the study found that women who were unaware of at least one cervical cancer patient, cervical cancer screening services and screened person were significantly associated with lower likelihood to screen for cervical cancer. These findings were consistent with studies conducted in China and Malta, which reported that unawareness of screening services was associated with lower uptake [34,35]. A study reported that knowing someone with cancer and/or screened was significantly associated with good knowledge and higher likelihood of screening [36]. Lack of adequate awareness and knowledge of cervical cancer screening services contributes to lower uptake. Awareness of peers who have utilized screening might influence others to uptake screening, as it might promote awareness of such key services. Similarly, the current study reported that women who had poor knowledge of cervical cancer's risk factors, signs, and symptoms, as

well as the groups at risk, were less likely to get screened. These findings concur with a study in Palestine, that reported poor knowledge of risk factors was associated with lower odds of screening [36]. This might be attributed to inadequate awareness campaigns and insufficient dissemination of information. Studies have reported that good knowledge of risk factors of a disease, signs and symptoms increase the likelihood of uptake of services to prevent and manage the disease [37–39]. Therefore, integration of regular and message tailored community-based health education campaigns might improve knowledge on risk factors and signs and symptoms. Hence, improved utilization of cervical cancer screening services among women.

## Conclusion

Cervical cancer screening is crucial for reducing incidence and mortality rates, but women in developing nations, particularly Kenya, have reported low utilization. The study revealed lower cervical cancer screening uptake compared to national prevalence, and the projected 70% screening uptake by 2030. This is due to insufficient information about cervical cancer and its screening, and inadequate promotional factors. Socio-demographic factors such as age, education attainment, employment status, household monthly income, family history and age at first marriage significantly influenced screening for cervical cancer. Similarly, awareness of cervical cancer, its screening process, and the outcomes for patients and screened individuals, along with knowledge of risk factors, signs, symptoms, and at-risk groups, significantly impacted the utilization of its screening. Thus, there is urgent need for intensified health education to enhance awareness and knowledge of women on cervical cancer and its screening.

## Supporting information

**S1 File. Data set underlying the study finding.**
(CSV)

## Acknowledgments

We gratefully acknowledge all the women who participated in this study for their time and cooperation. We would also like to thank the Kapkatet Sub-County Hospital administration and staff for their support and the research assistants for their dedicated efforts in data collection. This research was made possible with ethical approval from the University of Kabianga-Kapkatet Institutional Research Ethical Committee and the Kenyan National Commission for Science, Technology, and Innovation, whom we sincerely thank.

## Author contributions

**Conceptualization:** Joel Wanzala, Calvince Otieno Anino.

**Data curation:** Joel Wanzala, Calvince Otieno Anino.

**Formal analysis:** Joel Wanzala.

**Investigation:** Joel Wanzala, Calvince Otieno Anino.

**Methodology:** Joel Wanzala, Calvince Otieno Anino.

**Project administration:** Calvince Otieno Anino.

**Resources:** Joel Wanzala.

**Software:** Joel Wanzala, Calvince Otieno Anino.

**Supervision:** Joel Wanzala, Calvince Otieno Anino.

**Validation:** Joel Wanzala, Calvince Otieno Anino.

**Visualization:** Joel Wanzala, Calvince Otieno Anino.

**Writing – original draft:** Joel Wanzala.

**Writing – review & editing:** Joel Wanzala, Calvince Otieno Anino.

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
