## [Decision Letter · Decision Letter 0]

20 Aug 2025

Dear Dr. Wanzala,

Thank you for submitting your manuscript to PLOS ONE. After careful consideration, we feel that it has merit but does not fully meet PLOS ONE’s publication criteria as it currently stands. Therefore, we invite you to submit a revised version of the manuscript that addresses the points raised during the review process.

We look forward to receiving your revised manuscript.

Kind regards,

Prof . Lucy W. Kivuti-Bitok, Ph.D. MHSM,BScN

Academic Editor

PLOS ONE

Journal Requirements:

2. Please include captions for your Supporting Information files at the end of your manuscript, and update any in-text citations to match accordingly. Please see our Supporting Information guidelines for more information: http://journals.plos.org/plosone/s/supporting-information .

3. We note that there is identifying data in the Supporting Information file “CCS_Data[1].csv”. Due to the inclusion of these potentially identifying data, we have removed this file from your file inventory. Prior to sharing human research participant data, authors should consult with an ethics committee to ensure data are shared in accordance with participant consent and all applicable local laws.

-Location data

Reviewers' comments:

Reviewer's Responses to Questions

**Comments to the Author**

1. Is the manuscript technically sound, and do the data support the conclusions?

Reviewer #1: Yes

Reviewer #2: Yes

2. Has the statistical analysis been performed appropriately and rigorously?

Reviewer #1: Yes

Reviewer #2: No

3. Have the authors made all data underlying the findings in their manuscript fully available?

Reviewer #1: Yes

Reviewer #2: Yes

4. Is the manuscript presented in an intelligible fashion and written in standard English?

Reviewer #1: Yes

Reviewer #2: Yes

Reviewer #1: Abstract:

1. Consider rephrasing <of 85="" which=""> to 85% of which ….

Background

1. It is the leading cause of all cancer deaths in Kenya, with over 3,200 deaths in 2020 (2) -> this data is outdated. Consider more recent statistics and cite here

2. Most scholars -> consider changing to “experts”

3. This is alarming given that Kenya has been implementing a national screening program in the past 10 years. -> citation missing

4. Additionally, with the high level of awareness among Kenyan women reported by it is expected that uptake of cancer screening should be high -> grammar, check for additional /missing words here.

5. Thus, there could be other factors contributing to the low screening uptake. Therefore, this study aimed to assess the prevalence and factors associated with cervical cancer screening among women aged 18-49 years in Bureti Constituency, Kericho County. -> this sentence could be further strengthened to illustrate the gap in research

Study design

1. located in the rural setting. -> consider changing to : ….a rural setting

2. Sociodemographic section entailed 11 items asked questions -> (consider editing to): The sociodemographic section entailed 11 items (and) asked questions …

3. Consider editing “Those who scored a cut off of 50% and above were regarded to have good knowledge while those who failed were regarded poor.” -> Here’s a cleaner revision of your sentence: Those who scored 50% or above were considered to have good knowledge, while those who scored below 50% were considered to have poor knowledge.

4. Consider rephrasing: …to -> out of 328 respondents, 48 (33.1%) were aged 18–24 years; the majority were Christians (97.2%), married (50.3%), had attained secondary education (34.5%), reported business as their main source of income (39.3%), and earned less than Ksh. 2,000 (46.9%).

Association between sociodemographic factors and cervical cancer screening

1. “…..compared to those who aged 18–24 years.” -> consider rephrasing to compared to those aged 18–24 years

Discussion

1. Family history of cervical cancer is significantly -> .. “was significantly..”

2. According to Kenyan Demographic and Health Survey 2022 report reported that a national prevalence of cervical cancer screening of 16.7%, -> The Kenyan Demographic and Health Survey 2022 report showed a national…..

3. “symptoms of cervical cancer cervical cancer, and knowledge on groups at risk were less likely to screen” -> grammar error

4. Studies have reported that good knowledge of cervical cancer risk factors and signs and symptoms increase likelihood of screen uptak -> grammar

5. and groups at risk of cervical cancer significantly impact utilization its screening. T-> grammar</of>

Reviewer #2: General comments

The manuscript is presented clearly and concisely. The following are specific review comments, with the first two being minor and the last one being major.

Specific comments

1. The abstract results have to be presented uniformly. Some of the factors indicated toward the end do not have the respective OR and p-values like the prior variables.

2. The way the cross-tabulation tables were presented was misleading, and I would suggest that the authors do row percentages to identify the association of factors on the magnitude of screening.

3. Why are two models needed for the particular outcome variable? There should be a final model that takes into consideration variables significant in both models. The authors can organize them as models I, II, and III.

**Do you want your identity to be public for this peer review?** For information about this choice, including consent withdrawal, please see our Privacy Policy

Reviewer #1: No

Reviewer #2: No

---

## [Author Response · Author response to Decision Letter 1]

5 Sep 2025

All the reviewers' comments have been addressed as per the attached response to reviewers' letter.

---

## [Editor Report · Decision Letter 1]

17 Sep 2025

Dear Dr. Anino,

Thank you for submitting your manuscript to PLOS ONE. After careful consideration, we feel that it has merit but does not fully meet PLOS ONE’s publication criteria as it currently stands. Therefore, we invite you to submit a revised version of the manuscript that addresses the points raised during the review process.

We look forward to receiving your revised manuscript.

Kind regards,

Lucy W. Kivuti-Bitok, Ph.D. MHSM,BScN

Academic Editor

PLOS ONE

Journal Requirements:

Additional Editor Comments:

The renjoinder for this last feedback was not attached. Please attach or resend.

---

## [Author Response · Author response to Decision Letter 2]

23 Sep 2025

The correct rebuttal letter has been attached.

---

## [Decision Letter · Decision Letter 2]

4 Dec 2025

Uptake of Cervical Cancer Screening and Its Associated Factors Among Women of Reproductive Age in Kericho County

PONE-D-25-34672R2

Dear Dr. Anino,

We’re pleased to inform you that your manuscript has been judged scientifically suitable for publication and will be formally accepted for publication once it meets all outstanding technical requirements.

Kind regards,

Lucy W. Kivuti-Bitok, Ph.D. MHSM,BScN

Academic Editor

PLOS One

Additional Editor Comments (optional):

Reviewers' comments:

Reviewer's Responses to Questions

**Comments to the Author**

Reviewer #1: All comments have been addressed

2. Is the manuscript technically sound, and do the data support the conclusions?

Reviewer #1: Yes

3. Has the statistical analysis been performed appropriately and rigorously?

Reviewer #1: Yes

4. Have the authors made all data underlying the findings in their manuscript fully available?

Reviewer #1: Yes

5. Is the manuscript presented in an intelligible fashion and written in standard English?

Reviewer #1: Yes

Reviewer #1: Author addressed all reviewer comments as advised. The main issues were grammatical errors in all sections and presentation of results, which are adequately addressed

**Do you want your identity to be public for this peer review?** For information about this choice, including consent withdrawal, please see our Privacy Policy

Reviewer #1: No

---

## [Editor Report · Acceptance letter]

PONE-D-25-34672R2

PLOS One

Dear Dr. Anino,

I'm pleased to inform you that your manuscript has been deemed suitable for publication in PLOS One. Congratulations! Your manuscript is now being handed over to our production team.

Kind regards,

on behalf of

Prof Lucy W. Kivuti-Bitok

Academic Editor

PLOS One